

# 1 Transport of short-lived halocarbons to the
# 2 stratosphere over the Pacific Ocean.

Michal T. Filus[1], Elliot L. Atlas[2], Maria A. Navarro[2*], Elena Meneguz[3], David Thomson[3], Matthew J.
Ashfold[4], Lucy J. Carpenter[5], Stephen J. Andrews[5], Neil R.P. Harris[6]
1. Centre for Atmospheric Science, University of Cambridge, Cambridge, CB2 1EW, UK
2. Department of Atmospheric Sciences, RSMAS, University of Miami, Miami, Florida, USA
3. Met Office, Atmospheric Dispersion Group, FitzRoy Road, Exeter, EX1 3PB, UK
4. School of Environmental and Geographical Sciences, University of Nottingham Malaysia Campus,

10         43500,Semenyih, Selangor, Malaysia

5. Wolfson Atmospheric Chemistry Laboratories, Department of Chemistry, University of York,
York, YO10 5DD, UK
6. Centre for Environmental and Agricultural Informatics, Cranfield University, Cranfield, MK43
0AL, UK
*Correspondence to:* Neil Harris (neil.harris@cranfield.ac.uk)
**Abstract.** The effectiveness of transport of short-lived halocarbons to the upper troposphere and
lower stratosphere remains an important unknown in quantifying the supply of ozone-depleting
substances to the stratosphere. In early 2014, a major field campaign in Guam in the West Pacific,
involving UK and US research aircraft, sampled the tropical troposphere and lower stratosphere. The
resulting measurements of $CH_3I$, $CHBr_3$ and $CH_2Br_2$ are compared here with calculations from a
Lagrangian model. This methodology benefits from an updated convection scheme which improves
simulation of the effect of deep convective motions on particle distribution within the tropical
troposphere. We find that the observed $CH_3I$, $CHBr_3$ and $CH_2Br_2$ mixing ratios in the Tropical
Tropopause Layer (TTL) are consistent with those in the boundary layer when the new convection
scheme is used to account for convective transport. Particularly, comparisons between modelled
estimates and observations of shortest-lived $CH_3I$ indicates that the NAME convection scheme is
realistic up to the lower TTL but less good at reproducing the small number of extreme convective
events in the upper TTL. This study consolidates our understanding of the transport of short-lived
halocarbons to the upper troposphere and lower stratosphere by using improved model calculations
to confirm consistency between observations in the boundary layer, observations in the TTL, and
atmospheric transport processes. Our results support recent estimates of the contribution of short-
lived bromocarbons to the stratospheric bromine budget.

## 34 1 Introduction

The successful implementation of the Montreal Protocol with its adjustments and amendments has
led to reductions in stratospheric chlorine and bromine amounts since the late 1990s (Carpenter et al.,
2014). These reductions have halted the ozone decrease (Harris et al., 2015; Chipperfield et al., 2017;
Steinbrecht et al., 2017) with the exception of continued depletion in the lower stratosphere (Ball et
al., 2017). Recently, the importance of very short-lived (VSL) chlorine- and bromine containing

*Deceased: 19.12.2017



compounds has received a great deal of attention (e.g. Hossaini et al., 2017; Oram et al., 2017).
VSLS are not considered under the Montreal Protocol, but are required in order to ensure reconcile
between observed stratospheric measurements of inorganic or 'active' bromine with reported
anthropogenic bromine emission sources. VSLS input into the stratosphere has however remained a
poorly constrained quantity (Carpenter et al., 2014), which hinders our understanding of the on-going
decline in lower stratospheric ozone and our ability to make predictions of stratospheric ozone
recovery.
Three of the most important VSL halocarbons are $CH_3I$, $CHBr_3$ and $CH_2Br_2$. They have typical lower
tropospheric lifetimes (4, 15 and 94 days, respectively (Carpenter et al., 2014)) which are shorter
than tropospheric transport timescales and so they have non-uniform tropospheric abundances. They
are all emitted predominantly from the oceans and result principally from natural sources (e.g.
Lovelock 1975; Solomon et al., 1994; Oram and Penkett, 1994; Vogt et al., 1999; Salawitch et al.,
2006; Pyle et al., 2011; Carpenter et al., 2012, 2014; Tegtmeier et al., 2013; Saiz-Lopez et al., 2014).
The short-lived bromocarbons, chiefly $CHBr_3$ and $CH_2Br_2$, have been identified as the missing
source for the stratospheric active bromine (mostly originating from long-lived brominated organic
and inorganic substances; Pfeilsticker et al., 2000; Salawitch, 2006; Feng et al., 2007; Dessens et al.,
2009). The current estimates of the contribution of the short-lived bromocarbons to the active
bromine ($Br_y$) in the stratosphere range from 3-8 ppt (Liang et al., 2010, 2014; Carpenter et al., 2014;
Fernandez et al., 2014; Sala et al., 2014; Tegtmeier et al., 2015; Navarro et al., 2015, 2017; Hossaini
et al., 2016; Butler et al., 2017; Fiehn et al., 2017). Much of this uncertainty is linked to the
contribution of $CHBr_3$ which has both the shortest lifetime and the largest emissions of the
commonly observed bromocarbons.
The transport of VSL halocarbons into the lower stratosphere is by ascent through the tropical
tropopause layer (TTL) (Fueglistaler et al., 2009). An important factor influencing the loading of the
VSL bromocarbons in the TTL is the strength of the convective transport from the boundary layer
where the bromocarbons are emitted (Hosking et al., 2010; Russo et al., 2015; Fuhlbrügge et al.,
2016; Krzysztofiak et al., 2018). This is poorly quantified and, especially when taken together with
the large variations in boundary layer concentrations and the convection parameterisation being the
major source of uncertainty in chemistry transport models, limits our ability to model the bromine
budget in the current and future atmosphere (Liang et al., 2010, 2014; Hoyle et al., 2011; Russo et
al., 2011, 2015; Schofield et al., 2011; Aschmann et al., 2013; Fernandez et al., 2014; Hossaini et al.,
2016; Krzysztofiak et al., 2018).
To address this and other challenges, the Natural Environment Research Council Coordinated
Airborne Studies in the Tropics (NERC CAST), National Centre for Atmospheric Research
Convective Transport of Active Species in the Tropics (NCAR CONTRAST) and National
Aeronautics and Space Administration Airborne Tropical Tropopause Experiment (NASA
ATTREX) projects were organised (Harris et al., 2017; Jensen et al., 2017; Pan et al., 2017). These
projects joined forces in January-March 2014 in the American territory of Guam, in the West Pacific.
Three aircraft were deployed to sample air masses at different altitudes to investigate the



characteristics of the air masses affected by the deep convective systems. This campaign produced a
unique dataset of coordinated measurements for interpretative studies of transport and distribution of
the chemical species, including the VSL bromocarbons (Sect. 2.1 and 2.2). The NASA ATTREX
project also measured over the less convectively active east Pacific in January - February 2013.
The objective of this paper is to model the transport and distribution of $CH_3I$, $CHBr_3$ and $CH_2Br_2$ in
the TTL by quantifying their boundary layer and background contribution components using a new
Lagrangian methodology. Briefly, the approach quantifies how much of $CH_3I$, $CHBr_3$ and $CH_2Br_2$ in
the TTL come from the boundary layer, and assesses the role of convection in transporting these
compounds to the TTL. The calculation is completed by estimating the background component (i.e.
how much of $CH_3I$, $CHBr_3$ and $CH_2Br_2$ originate from outside the immediate boundary layer source).
Section 2 presents an overview of the field campaigns, the $CH_3I$, $CHBr_3$ and $CH_2Br_2$ measurements,
and how the NAME calculations are used. In Section 3, the approach is illustrated by comparing
model estimates and measurements from one ATTREX 2014 flight. This analysis is then expanded to
cover measurements from all ATTREX 2014 and 2013 flights. The role of convection in transporting
VSL halocarbons to the TTL is further examined in Section 4. Based on the modelled calculations of
$CHBr_3$ and $CH_2Br_2$, Section 5 discusses how much these VSL bromocarbons contribute to the
bromine budget in the TTL.

## 2 Methodology

### 2.1 Overview of the CAST, CONTRAST and ATTREX campaigns

The joint CAST, CONTRAST and the third stage of the ATTREX campaign took place in January-
March 2014, in the West Pacific. Guam ($144.5^o$ E, $13.5^o$ N) was used as a research mission centre for
these three campaigns. Three aircraft were deployed to measure physical characteristics and chemical
composition of tropical air masses from the earth's surface up to the stratosphere. In CAST, the
Facility for Airborne Atmospheric Measurements (FAAM) BAe-146 surveyed the boundary layer
and lower troposphere to sample the convection air mass inflow, while in CONTRAST the National
Science Foundation - National Center for Atmospheric Research (NSF-NCAR) Gulfstream V (GV)
principally target the region of maximum convective outflow in the mid- and upper troposphere and
also sampled the boundary layer. Finally, in ATTREX, the NASA Global Hawk (GH) sampled the
TTL to cover air masses likely to be detrained from the higher convective outflow. For more details
on these campaigns, in particular, objectives, meteorological conditions and descriptions of
individual flights, please refer to the campaign summary papers: Harris et al., 2017 (CAST), Pan et
al., 2017 (CONTRAST) and Jensen et al., 2017 (ATTREX). ATTREX had four active measurement
campaigns, and we also consider the second campaign which was based in Los Angeles in January-
March 2013 and which extensively sampled the East and Central Pacific TTL in six research flights.

### 2.2 Measurements of the VSL halocarbons

Whole Air Samplers (WAS) were deployed on all three aircraft to measure VSL halocarbons. The
FAAM BAe-146 and NSF-NCAR GV also used on-board gas chromatography-mass spectrometry



(GC-MS) system for real-time analysis (Wang et al., 2015; Andrews et al., 2016; Pan et al., 2017),
though these measurements are not used in our analysis. WAS instrumentation had been used
routinely in previous deployments. The sampling and analytical procedures are capable of accessing
a wide range of mixing ratios at sufficient precision and the measurements from the three aircraft
have been shown to be consistent and comparable (Schauffler et al., 1998; Park et al., 2010; Andrews
et al., 2016).
The CAST VSL halocarbon measurements were made using the standard FAAM WAS canisters
with 30 second filling time. Up to 64 samples could be collected on each flight and these were
analysed in the aircraft hangar, usually within 72 hours after collection. Two litres of sample air were
pre-concentrated using a thermal desorption unit (Markes) and analysed with GC-MS (Agilent 7890
GC, 5977 Xtr MSD). Halocarbons were quantified using a NOAA calibration gas standard. The
measurement and calibration technique is further described and assessed in Andrews et al. (2013;
128    2016).

The ATTREX AWAS sampler consisted of 90 canisters, being fully automated and controlled from
the ground. Sample collection for the AWAS samples was determined on a real-time basis depending
on the flight plan altitude, geographic location, or other relevant real-time measurements. The filling
time for each canister ranged from about 25 seconds at 14 km to 90 seconds at 18 km. Canisters were
immediately analysed in the field using a high performance GC-MS coupled with a highly sensitive
electron capture detector. The limits of detection are compound-dependent and vary from ppt to sub-
ppt scale, set at 0.01 ppt for $CHBr_3$, $CH_2Br_2$ and $CH_3I$ (Navarro et al., 2015). A small artefact of
~0.01-0.02 ppt for $CH_3I$ cannot be excluded. AWAS samples collected on the GV were analysed
with the same equipment. Detailed comparison of measurements from the three systems found
agreement within ~7 % for $CHBr_3$, ~3 % for $CH_2Br_2$, and 15 % for $CH_3I$ (Andrews et al., 2016).
**2.3 UK Meteorological Office NAME Lagrangian Particle Dispersion Model**
The Lagrangian particle dispersion model, NAME, (Jones, et al., 2007) is used to simulate the
transport of air masses in the Pacific troposphere and the TTL. Back trajectories are calculated with
particles being moved through the model atmosphere by mean wind fields (0.352° longitude and
0.235° latitude, i.e. ~25 km, with 31 vertical levels below 19 km) calculated by the Meteorological
Office's Unified Model at 3-hour intervals. This is supplemented by a random walk turbulence
scheme (Davies et al., 2005). For this analysis, the NAME model is used with the improved
convection scheme, (Meneguz and Thomson, 2014) which simulates displacement of particles
subject to convective motions more realistically than previously (Meneguz et al., in review). NAME
is run backward in time to determine the origin(s) of air measured at a particular location (WAS
sample) along the ATTREX GH flight track.
15,000 particles are released from each point along the flight track where VSL halocarbons were
measured in WAS samples. To initialise the NAME model, particles are released randomly in a
volume with dimensions 0.1° × 0.1° × 0.3 km centred on each sample. As particles are followed 12





days back in time, trajectories are filtered on the basis of first crossing into the boundary layer (1
km). Subsequently, the fraction of particles which crossed below 1 km is calculated for each WAS
measurement point (Ashfold et al., 2012). The NAME 1 km fractions are indicative of the boundary
layer air mass influence to the TTL. The 1 km boundary layer fractions are then used to
quantitatively estimate the VSL halocarbon contribution to the TTL from the boundary layer,
$[X]_{BL\_Contribution}$. In order to compare the measured and modelled halocarbon values, estimates of the
contribution from the background troposphere, $[X]_{BG\_Contribution}$ (i.e. air which has not come from the
boundary layer within 12 days) are made. The model estimate for the total halocarbon mixing ratio,
$[X]_{NAME\_TTL}$, is thus given by Eq. (1):

$$[X]_{NAME_{TTL}} = [X]_{BL\_Contribution} + [X]_{BG\_Contribution} \qquad (1)$$

The methods for calculating $[X]_{BL\_Contribution}$ and $[X]_{BG\_Contribution}$ are now described.

**2.3.1 NAME modelled boundary layer contribution**

The contribution from the boundary layer, ($[X]_{BL\_Contribution}$ - described above) to the VSLs in the TTL
can be estimated using
(i)    the fractions of trajectories crossing below 1 km in the previous 12 days;
(ii)    the transport times to the TTL calculated for each particle;
(iii)    the initial concentration values for $CH_3I$, $CHBr_3$ and $CH_2Br_2$; and
(iv)    their atmospheric lifetimes (to account for the photochemical removal along the trajectory).
More specifically, the boundary layer contribution to the TTL for the VSL halocarbons is calculated
using Eq. (2) and Eq. (3):

$$[X]_{BL_{Contribution},t} = [X]_{BL} \times fraction_t \times exp^{(-t/\tau)} \qquad (2)$$

$$[X]_{BL\_Contribution} = \Sigma\left([X]_{BL_{Contribution},t}\right) \qquad (3)$$

Equation (2) gives the boundary layer contribution to the TTL for a given tracer, X (where X could
be $CH_3I$, $CHBr_3$, $CH_2Br_2$), at model output time step, t. The model output time step used is 6 hours,
from t = 0 (particle release) to t = 48 (end of a 12 day run). $[X]_{BL}$ stands for the initial boundary layer
concentration of a given tracer - assigned to each particle which crossed below 1 km (Table 1).
Fraction$_t$ is a number of particles which first crossed 1 km in a model output time step, t, over a total
number of particles released, and $exp^{(-t/\tau)}$ is a term for the photochemical loss (where $\tau$ stands for
atmospheric lifetime of a respective VSL halocarbon). Equation (3) gives the boundary layer
contribution that is the sum of boundary layer contribution components in all model output time steps
(for t = 1 to 48).
Equation (2) calculates the decay of each tracer after it leaves the boundary layer (0-1 km) which is
valid for a well-mixed boundary layer. Since 15,000 particles are released for each AWAS sample,





contributions from each particle from below 1 km in the previous 12 days are summed. Decay times,
$\tau$, of 4, 15 and 94 days for $CH_3I$, $CHBr_3$ and $CH_2Br_2$, respectively, are used (i.e. constant chemical
loss rate) (Carpenter et al., 2014). Thus, a particle getting to the TTL in 1 day contributes more of a
given tracer to that air mass than a particle taking 10 days. Once this chemical loss term was taken
into account, the NAME trajectories can be used to calculate the contribution of convection of air
masses from the boundary layer within the preceding 12 days. The initial boundary layer
concentrations are derived from the CAST and CONTRAST WAS measurements taken in the West
Pacific in the same period of January-March 2014 as for the ATTREX measurements in the TTL
(Table 1). These observed means are used in model calculations, and the similarity between them and
literature values reported in Carpenter et al. (2014) is seen, with lower values for $CHBr_3$ only.
**2.3.2 NAME modelled background contribution**
To compare our model results against the AWAS observations, the background contribution,
$[X]_{BG\_Contribution}$ (meaning the contribution from the fraction of trajectories which do not cross below 1
km within 12 days) also needs to be accounted for. This requires estimates for the fraction of
trajectories from the free troposphere, which is (1-fraction$_{BL}$) , Eq. (4), and an estimate of the
halocarbon mixing ratio in that fraction, $[X]_{BG}$, Eq. (5) i.e.
$$fraction_{BL} = \sum(fraction_t) \tag{4}$$
$$[X]_{BG\_Contribution} = (1 - fraction_{BL}) \times [X]_{BG} \tag{5}$$
Since each sample has 15,000 back-trajectories associated with it, some of which came from below 1
km and some of which did not, a definition as to which air samples are considered as boundary layer
and which are considered background is required. Two approaches are tested. Both use the NAME
calculations to identify AWAS samples in all flights (2013 and 2014) with low convective influence
by (i) filtering for air masses with boundary layer fraction values less than 1, 5 or 10 %; and (ii)
selecting the lowest 10 % of boundary layer fractions. Then, the $CH_3I$, $CHBr_3$ and $CH_2Br_2$ AWAS
observations, corresponding to the boundary layer fraction values less than 1, 5 or 10 %, or the
lowest 10 % of boundary layer fractions, are averaged to provide $CH_3I$, $CHBr_3$ and $CH_2Br_2$
background mixing ratios. Two approaches are explored below (Sect. 3.1.2).
**3 Analysis of ATTREX 2014 Research Flight 02**
We start by showing our results from one of the individual ATTREX 2014 Research Flights, RF02,
to illustrate the method. This is followed by analysing all Research Flights together for ATTREX
2014 and 2013 in Sect. 4, and calculating the modelled contribution of active bromine from very-
short lived brominated substances, $CHBr_3$ and $CH_2Br_2$, to the TTL (Sect. 5).
**3.1 Individual ATTREX 2014 Flight: Research Flight 02**





Figure 1 shows the vertical distribution of $CH_3I$, $CHBr_3$ and $CH_2Br_2$ in the TTL observed during the
individual research flight, RF02, during ATTREX 2014. Held on 16-17 February 2014, RF02 was
conducted in a confined area east of Guam (12-14° N, 145-147° E) due to a faulty primary satellite
communications system for Global Hawk command and control (Jensen, et al., 2017). 26 vertical
profiles through TTL were made, with 86 AWAS measurements taken in total. A high degree of
variability of $CH_3I$ in the TTL was observed (from > 0.4 ppt at 14-15 km, to near-zero ppt values at
17-18 km). Each profile, in general, showed a gradation in $CH_3I$ distribution in the TTL. Higher
values were measured in the lower TTL up to 16 km, with values decreasing with altitude. The same
pattern was observed for $CHBr_3$ and $CH_2Br_2$, with the highest concentrations measured in the lower
TTL (14-15 km), and the lowest at 17-18 km.
**3.1.1 NAME modelled boundary layer contribution**
Figure 2(a) shows the vertical distribution of the boundary layer air contribution to the TTL
(corresponding to the AWAS measurement locations along the RF02 flight track). It reveals higher
boundary layer air influence in the lower TTL, decreasing with altitude (similarly to the VSL
halocarbon observations). Cumulatively, the highest fractions from below 1 km are found for the
lower TTL (14-15 km). A noticeable decrease occurs between the lower and upper TTL (15 to 17
km). From 16 km up, little influence (indicated by <10 % and <5 % 1 km fractions of trajectories
below 1 km for 16-17 km and 17-18 km, respectively) of the low-level air masses is seen.
Figure 2(b) shows all NAME runs for RF02 grouped into four 1 km TTL bins: 14-15 km, 15-16 km,
16-17 km and 17-18 km. In the 14-15 km bin, most particles from the low troposphere are calculated
to have arrived in the preceding 4 days with many in the preceding 2 days. This represents the fast
vertical uplift of the low tropospheric air masses to the lower TTL. At 15-16 km, two particle
populations are observed: the first group results from recent vertical uplift, while the second group
has been in the upper troposphere for longer than a couple of days (see Fig. 2c in Navarro et al., 2015
for similar example). Above 16 km, the overwhelming majority (>90 %) of the released particles are
calculated to be in the TTL for the previous 12 days, with negligible evidence for transport from the
low troposphere. This shows the dominance of the long-range, horizontal transport for the 16-17 and
17-18 km NAME runs (also shown in Navarro et al., 2015).
Figure 3 shows the locations at which trajectories crossed 1 km, thereby indicating boundary layer
source regions for the RF02 TTL air masses. Boundary layer sources in the western and central
Pacific are the most important for the lowest TTL bin (14-15 km, Fig. 3a) in this flight. The Maritime
Continent, the Northern Australia coast, the Indian Ocean and the equatorial band of the African
continent increase in importance as altitude increases, though the overall contribution of recent
boundary layer air masses decreases with increasing altitude.
Figure 4 shows the NAME modelled boundary layer contribution to the TTL for $CH_3I$, $CHBr_3$ and
$CH_2Br_2$ during RF02. It is important to note that this contribution corresponds to uplift from below 1
km in the preceding 12 days, the length of the trajectories. The calculated boundary layer
contributions for $CH_3I$, $CHBr_3$ and $CH_2Br_2$ from the 1 km fractions are highest at 14-15 km,





dropping off with altitude. Almost no boundary layer contribution is found for 17-18 km (with values
close to 0 ppt).

### 3.1.2 NAME modelled background contribution

Here we explore the two approaches described in Sect. 2.3.2 for estimating the $CHBr_3$ and $CH_2Br_2$
background mixing ratios. Similar values are seen in ATTREX 2013 and 2014. Less variation is
observed for $CH_2Br_2$ due to its longer atmospheric lifetime.
ATTREX 2013 and 2014 are treated separately in the analysis presented below due to the difference
in $CH_3I$ background estimates. The approach using the lowest 10 % of the boundary layer fractions is
used to estimate the background contribution for the 2014 flights as not enough data meet the former
condition due to the proximity of the flights to strong convection. The background values, inferred
from all the ATTREX 2014 flights, are used in the individual flight calculations as again there are
not enough data from an individual flight to make background calculations for that flight. In
ATTREX 2013 we use the boundary layer fractions less than 5 % approach for the $CH_3I$ background
estimation. The ATTREX 2014 background estimates should be taken as upper limits as it is hard to
identify samples with no convective influence in 2014. This is especially true for the lower TTL
since the ATTREX 2014 flights were close to the region of strong convection.
Figure 5 shows the VSL background mixing ratios calculated for the ATTREX campaigns in 2013
and 2014. In ATTREX 2013, low $CH_3I$ background mixing ratios are found. All approaches show
similar background mixing ratios. In 2014, higher $CH_3I$ background mixing ratios are calculated due
to ubiquity of air from recent, vertical uplift. No boundary layer fractions less than 1 % are found for
the 14-17 km bins, and less than 5 % for the 14-15 km.

### 3.1.3 NAME modelled total concentrations

The NAME boundary layer and background contribution estimates are added to give an estimate for
total halocarbon mixing ratio, $[X]_{NAME\_TTL,}$ (Eq. (1)), for comparison with the AWAS observations.
Figure 6 and Table 2 show the vertical distribution of NAME-based estimates for $CH_3I$, $CHBr_3$ and
$CH_2Br_2$ in the TTL for RF02. The sums of the NAME $CH_3I$, $CHBr_3$ and $CH_2Br_2$ boundary layer and
background contribution estimates agree well with the AWAS observations for all the 1 km TTL bins
(compared with Fig. 1).
At 14-15 km, the modelled boundary layer contribution of $CH_3I$ is similar to the observations,
indicating recent, rapid convective uplift. This provides evidence that the improved convection
scheme provides a realistic representation of particle displacement via deep convection. At higher
altitudes, the background contribution is more important and, indeed, the modelled total $CH_3I$ values
are greater than the observations. This overestimate of the background contribution results from the
difficulty of identifying samples with no convective influence in ATTREX 2014. This problem is
most important for $CH_3I$ with its very short lifetime.



CHBr$_3$ drops off slower with altitude than CH$_3$I and quicker than CH$_2$Br$_2$. At 14-15 km, the boundary
layer contribution accounts for ~ 50 % of the modelled sums of CHBr$_3$ and CH$_2$Br$_2$, but less than 5 %
for CHBr$_3$ and CH$_2$Br$_2$ at 17-18 km. For the upper TTL, the background contribution estimates
constitute over 85 % of the modelled sums, thus taking on more importance.

**4 The role of transport in the VSL halocarbon distribution in the TTL**

The role of transport in the CH$_3$I, CHBr$_3$ and CH$_2$Br$_2$ distribution in the TTL is examined in this
section by applying the NAME based analysis introduced in Sect. 3 to all CH$_3$I, CHBr$_3$ and CH$_2$Br$_2$
AWAS observations in the ATTREX 2013 and 2014 campaigns.
In ATTREX 2013, six flights surveyed the East Pacific TTL in February-March 2013. Four flights
went west from Dryden Flight Research Centre to the area south of Hawaii, reaching 180$^o$ longitude.
Little influence of convective activity was observed. Most samples with strong boundary layer
influence were observed in air masses that had originated over the West Pacific and the Maritime
Continent, where it was uplifted to the TTL and transported horizontally within the TTL (Navarro et
al., 2015). Two flights sampled the TTL near the Central and South American coast. Few convective
episodes were observed. The sampled air had predominantly a small boundary layer air signature
from the West Pacific and the Maritime Continent.
In ATTREX 2014, two transit flights and six research flights were made in the West Pacific in
January-February 2014. This period coincided with the active phase of Madden-Julian Oscillation
(MJO) and increased activity of tropical cyclones. A large influence of recent convective events is
observed (Navarro et al., 2015), reflected in the elevated CH$_3$I and CHBr$_3$ mixing ratios and the high
values of NAME fractions of trajectories below 1 km. All three aircraft flew together in 2014 and so
there is a more complete set of measurements from the ground up. Accordingly, this year is discussed
first.

**4.1 VSL halocarbon distribution in the TTL: ATTREX 2014**

Figure 7 shows the vertical distribution of the observations and of the modelled boundary layer
contribution and total mixing ratios for CH$_3$I, CHBr$_3$ and CH$_2$Br$_2$ for all the ATTREX 2014 flights
(using only the AWAS measurements made from 20$^o$N southward). As in RF02, CH$_3$I is highest in
the lower TTL, dropping off with altitude. Large flight-to-flight variability in CH$_3$I measurements is
seen. The fraction of NAME particles that travel below 1 km in the previous 12 days (Table 3) are
highest at 14-15 km (mean of 57 %) and decrease with altitude in a similar fashion. The CH$_3$I
boundary layer contribution explains most of the observations for the 14-15 and 15-16 km layers.
Disparities in observed and modelled CH$_3$I arise from 16 km up. Background estimate values are
minimal, oscillating between 0 and the limit of detection of the AWAS instrument for the iodinated
short-lived organic substances, 0.01 ppt. The sums of the CH$_3$I boundary layer and background



contribution estimates show good agreement with AWAS observations for all the TTL 1 km
segments (Table 3).
The good agreement for the 14-15 km and 15-16 km layers can be attributed to the improved
representation of deep convection in NAME, provided by the new convection scheme (Meneguz et
al., in review). However, there is an underestimation of the boundary layer contribution to the upper
TTL levels (16-17 and 17-18 km) which we attribute to the new convection scheme not working as
well at these altitudes. Both the $CH_3I$ AWAS observations and the modelled sums are higher than
reported previously in the literature (Carpenter et al., 2014) for all the TTL segments. This may be
explained by sampling the TTL in a region of high convective activity. This result gives confidence
in the quality of the new convection scheme and hence in similar calculations of convective influence
on the longer-lived $CHBr_3$ and $CH_2Br_2$.
The highest $CHBr_3$ and $CH_2Br_2$ concentrations were observed in the lower TTL (14-15 km),
dropping off more slowly with altitude than $CH_3I$. The weight of the modelled boundary layer
contribution estimates to the modelled total amounts varies from approximately 50% at 14-15 km
(unlike for $CH_3I$ where over 85 % of the modelled sum is attributed to the boundary layer
contribution at 14-15 km) to < 20% at 17-18 km. The sums of the boundary layer and background
contribution estimates show good agreement with $CHBr_3$ and $CH_2Br_2$ AWAS observations. The
ATTREX observations and the NAME modelled sums are within the range of values reported in the
literature (Carpenter, et al., 2014).
**4.2 VSL halocarbon distribution in the TTL: ATTREX 2013**
Figure 8 shows the vertical distribution for $CH_3I$, $CHBr_3$ and $CH_2Br_2$ in the TTL, observed and
modelled from the ATTREX 2013 flights (using only the AWAS measurements taken south of
20$^o$N). Much lower $CH_3I$ values are found in 2013 than in 2014 (Fig. 7). The NAME 1 km fractions
are considerably lower (~fourfold), and the corresponding $CH_3I$ boundary layer contribution shows
values close to the limit of detection of the AWAS instrument for $CH_3I$. The background contribution
comprises over 85-90 % of the sums of the modelled $CH_3I$ estimate in the TTL. Good agreement is
found between the sums of the boundary layer and background contribution estimates, against the
AWAS observations. Both the observed and modelled values are in the low end of the $CH_3I$
concentrations reported by the WMO 2014 Ozone Assessment (Carpenter et al., 2014).
The ATTREX 2013 mixing ratios are also lower for $CHBr_3$ and higher $CH_2Br_2$ than shown in Fig. 7
for 2014. The NAME calculated $CHBr_3$ and $CH_2Br_2$ boundary layer contributions are small,
constituting approximately 10 % of the NAME modelled sums for 14-15 km, and less for the upper
TTL segments. The background contribution estimates comprise over 85 % of the modelled sums.
Good agreement is found between the sums of the boundary layer and background contribution
estimates and the $CHBr_3$ and $CH_2Br_2$ AWAS observations.
**4.3 ATTREX 2013 and 2014: Inter-campaign comparison**



Clear differences in the vertical distributions of $CH_3I$ in the TTL are found in ATTREX 2013 and
2014. $CH_3I$ estimates, corresponding to high values in the NAME modelled 1 km fractions, are high
in 2014, whereas in 2013 almost no $CH_3I$ is estimated to be in the TTL. This is due to the minimal
contribution of the boundary layer air within the previous 12 days: ATTREX 2013 was in the East
Pacific away from the main region of strong convection. Longer transport timescales result from
horizontal transport and were more important in ATTREX 2013, with much less recent convective
influence than in ATTREX 2014. More chemical removal of $CH_3I$ and $CHBr_3$ thus took place,
leading to lower concentrations in the East Pacific TTL.
The trajectories are analysed to investigate the timescales for vertical transport by calculating how
long it took particles to go from below 1 km to the TTL. In 2013, almost no episodes of recent rapid
vertical uplift are found, with most particles taking 8 days and more to cross the 1 km. This is
indicative of the dominant role of long-range horizontal transport. In 2014, by way of contrast, a
considerable number of trajectories (10's of per cent) come from below 1 km in less than 4 days,
representing the 'young' air masses being brought from the low troposphere via recent and rapid
vertical uplift.
The spatial variability in the boundary layer air source origins, as well as the variation in atmospheric
transport pathways and transport timescales can explain the differences in the distribution of the
NAME 1 km fractions in the TTL. In 2014 (2013), higher (lower) boundary layer fractions
corresponded well with higher (lower) $CH_3I$ and $CHBr_3$ values in the TTL, especially with the
highest concentrations occurring for the flights with the most convective influence and the highest
fractions of particles arriving within the 4 days.
In the ATTREX 2014 flights, the western and central Pacific is the dominant source origin of
boundary layer air to the TTL (Navarro et al., 2015). Increased tropical cyclone activity in this area
(particularly Faxai 28 February – 6 March 2014 and Lusi 7-17 March 2014) and the strong signal
from the MJO related convection contributed to the more frequent episodes of strong and rapid
vertical uplifts of the low-level air to the TTL. A significant contribution is also seen from the central
Indian Ocean, marking the activity of the Fobane tropical cyclone (6-14 February 2014). Minimal
contribution from the other remote sources (Indian Ocean, African continental tropical band) is found
(Anderson et al., 2016; Jensen et al., 2017; Newton et al., 2018).
**5 How much do VSL bromocarbons contribute to the bromine budget in the TTL?**
The NAME modelled $CHBr_3$ and $CH_2Br_2$ estimates in the TTL are used to calculate how much
bromine from the VSL bromocarbons, $Br\text{-}VSL_{org}$, is found in the lower stratosphere, based on how
much enters the TTL in the form of bromocarbons (as in Navarro et al. (2015)). $CHBr_3$ and $CH_2Br_2$
are the dominant short-lived organic bromocarbons, and the minor bromocarbons: $CH_2BrCl$,
$CHBr_2Cl$ and $CHBrCl_2$ are excluded here (as their combined contribution is less than 1 ppt to $Br\text{-}$
$VSL_{org}$ at 14-18 km, Navarro et al., 2015). The NAME modelled $CHBr_3$ and $CH_2Br_2$ estimates are
multiplied by the number of bromine atoms (bromine atomicity), and then summed to yield the total
of $Br\text{-}VSL_{org}$.



Figure 9 shows the contribution of $CHBr_3$ and $CH_2Br_2$, the two major VSL bromocarbons
contributing to the bromine budget in the TTL. For ATTREX 2013 and 2014, similar contributions
of $CHBr_3$ and $CH_2Br_2$ to Br-$VSL_{org}$ are found in the lower TTL. In 2014, $CHBr_3$ in the lower TTL
was abundant enough to contribute as much Br-$VSL_{org}$ as $CH_2Br_2$. A combination of larger boundary
layer air influence in the TTL and shorter mean transport times to reach the TTL result in the
observed higher $CHBr_3$ contribution to the Br-$VSL_{org}$ in the lower TTL in 2014, than in 2013. The
$CH_2Br_2$ contribution dominates in the upper TTL due to its longer atmospheric lifetime.
Good agreement is found between the bromine loading from the VSL bromocarbons, inferred from
the NAME modelled estimates initialised with BAe-146 and GV measurements, and the Global
Hawk AWAS observations. Higher organic bromine loading is seen around the cold point tropopause
(16-17 km) in ATTREX 2014.
Using the upper troposphere measurements taken during the SHIVA campaign in the western Pacific
in November-December 2011, Sala et al. (2014) calculated an estimate for VSLS ($CHBr_3$, $CH_2Br_2$,
$CHBrCl_2$, $CH_2BrCl$, $CHBr_2Cl$) contribution to the organic bromine at the level of zero radiative
heating (15.0 - 15.6 km). Air masses reaching this level are expected to reach the stratosphere. This
VSLS mean mixing ratio estimate of 2.88 (+/- 0.29) ppt (2.35 ppt for $CHBr_3$ and $CH_2Br_2$, excluding
minor short-lived bromocarbons) is lower due to a lower contribution from $CHBr_3$ estimate (0.22 ppt
compared to $CHBr_3$ estimate for NAME / ATTREX in Table 5). Compared to other literature values
reported in Sala et al., (2014), our estimates of the contribution of $CHBr_3$ and $CH_2Br_2$ to the organic
bromine at the LZRH are slightly higher largely due to a higher estimate for a shorter-lived $CHBr_3$.
Navarro et al. (2015) report slightly higher bromine loading from the Br-$VSL_{org}$ at the tropopause
level (17 km) in the West Pacific, 2014 than in the East Pacific, 2013 (the Br-$VSL_{org}$ values from the
AWAS observations were of 3.27 (+/-0.47) and 2.96 (+/-0.42) ppt, respectively). The minor short-
lived organic bromine substances were included in the analysis of Navarro et al. (2015), accounting
for the higher Br-$VSL_{org}$.
Butler et al. (2017), report a mean mole fraction and range of 0.46 (0.13-0.72) ppt and 0.88 (0.71-
1.01) ppt of $CHBr_3$ and $CH_2Br_2$ being transported to the TTL during January and February 2014.
This is consistent with a contribution of 3.14 (1.81-4.18) ppt of organic bromine to the TTL over the
region of the campaign. The NAME modelled results presented here (Fig. 9, Table 5) are in good
agreement with the values reported by Navarro et al. (2015) and Butler et al. (2017).
**6 Summary and Discussion**
We have used the NAME trajectory model in backward mode to assess the contribution of recent
convection to the mixing ratios of three short-lived halocarbons, $CH_3I$, $CHBr_3$ and $CH_2Br_2$. 15,000
back-trajectories are computed for each measurement made with the whole air samples on the NASA
Global Hawk in ATTREX 2013 and 2014, and the fraction that originated below 1 km is calculated.
A steep drop-off in this fraction is observed between 14-15 km and 17-18 km. Low level
measurements of $CH_3I$, $CHBr_3$ and $CH_2Br_2$ from the FAAM BAe-146 and the NCAR GV are used in





conjunction with these trajectories and an assumed photochemical decay time to provide estimates of
the amount of each gas reaching the TTL from below 1 km. Comparison of these modelled estimates
with the $CH_3I$ measurements shows good agreement with the observations at the lower altitudes in
the TTL values, with less good agreement at altitudes > 16 km, though it should be noted that the
amounts are very small here. The lifetime of $CH_3I$ is 3-5 days, and so there is a > 90 % decay in the
12 day trajectories. The comparison between the modelled and measured $CH_3I$ thus indicates that the
NAME convection scheme is realistic up to the lower TTL but less good at reproducing the small
number of extreme convective events that penetrate to the upper TTL.
In order to perform similar calculations for the longer-lived bromocarbons, an estimate of the
background free tropospheric concentration is required. This is calculated by considering
bromocarbon values in samples where there was only a small influence from the boundary layer, i.e.
where very few NAME trajectories passed below 1 km. This is possible in 2013 when the ATTREX
flights were away from the region of strong convection, but much harder in 2014 when (as planned)
the flights were heavily influenced by convection. By summing the boundary layer and background
contributions, an estimate of the total bromocarbon mixing ratio is obtained.
The resulting modelled estimates are found to be in generally good agreement with the ATTREX
measurements. In other words, a high degree of consistency is found between the low altitude
halocarbon measurements made on the BAe-146 and GV and the high altitude measurements made
on the Global Hawk when they are connected using trajectories calculated by the NAME dispersion
model with its updated convection scheme and driven by meteorological analyses with 25 km
horizontal resolution.
In the above, the boundary layer contribution arises from trajectories which visit the boundary layer
within 12 days while the background contribution involves air that has been transported into the TTL
from outside the boundary layer on timescales up to 12 days. Sensitivity tests were performed in
which the trajectories were followed for longer than 12 days: the effect was to re-allocate some of the
air from the background category into the boundary layer contribution with no net change in the
total.
The approach using NAME trajectories and boundary layer measurements produces Br-VSL$_{org}$
estimates of 3.47 +/- 0.4 (3.3 +/- 0.4) ppt in the lower East (West) Pacific TTL (14-15 km) and 2.5
+/-0.2 (2.4 +/-0.4) ppt in the upper East (West) Pacific TTL (17-18 km). These lie well within the
range of the recent literature findings (Tegtmeier et al., 2012; Carpenter et al., 2014; Liang et al.,
2014; Navarro et al., 2015; Butler et al., 2017). The validation with the ATTREX measurements
provides confidence that a similar approach could be used for years when high altitude measurements
are not available assuming that realistic estimates of the background tropospheric contributions can
be obtained from either models or measurements.
**7 Data availability**




The CH$_3$I, CHBr$_3$ and CH$_2$Br$_2$ AWAS data from the NASA ATTREX measurements are available online in the NASA ATTREX database (https://espoarchive.nasa.gov/archive/browse/attrex). The CAST measurements are stored on the British Atmospheric Data Centre, which is part of the Centre for Environmental Data archive at http://catalogue.ceda.ac.uk/uuid/565b6bb5a0535b438ad2fae4c852e1b3. The CONTRAST AWAS data are available through http://catalog.eol.ucar.edu/contrast. The NAME data are available from the corresponding author upon request.

## 8 Author Contribution

The main part of the analysis was conducted by MF. EA and MN provided CH$_3$I, CHBr$_3$ and CH$_2$Br$_2$ AWAS measurements from the ATTREX and CONTRAST research flights. SA and LC provided CH$_3$I, CHBr$_3$ and CH$_2$Br$_2$ measurements from the CAST campaign. MA designed initial scripts for NAME runs and products. EM and DT developed the model code for improved convection scheme. MF and NH prepared the manuscript with contributions from all co-authors, NH also supervised this PhD work.

## 9 Acknowledgements

The authors would like to thank our NASA ATTREX, NCAR CONTRAST and NERC CAST project partners and the technical teams. MF would like to thank Drs Michelle Cain, Alex Archibald, Sarah Connors, Maria Russo and Paul Griffiths for their input on the NAME applications for flight planning and post-flight modelling. The research was funded through the UK Natural Environment Research Council CAST project (NE/J006246/1 and NE/J00619X/1), and MF was supported by a NERC PhD studentship. EA acknowledges support from NASA grants NNX17AE43G, NNX13AH20G and NNX10AOB3A. We acknowledge use of the NAME atmospheric dispersion model and associated NWP meteorological datasets made available to us by the UK Met Office.

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

**11 Tables**
*Table 1. Boundary layer concentrations and atmospheric lifetimes for $CH_3I$, $CHBr_3$ and $CH_2Br_2$*
*(Carpenter et.al., 2014).*

| Tracer, [X] | Boundary Layer Concentration, $[X]_{BL}$ [ppt] | | Atmospheric Lifetime, $\tau$ [days] |
|---|---|---|---|
| | CAST and CONTRAST<br><br>Mean (Range) Median | Carpenter et al., 2014<br><br>Median (Range) | |
| $CH_3I$ | 0.70 (0.16-3.34) 0.65 | 0.8 (0.3-2.1) | 4 |
| $CHBr_3$ | 0.83 (0.41-2.56) 0.73 | 1.6 (0.5-2.4) | 15 |
| $CH_2Br_2$ | 0.90 (0.61-1.38) 0.86 | 1.1 (0.7-1.5) | 94 |





*Table 2. ATTREX 2014 Research Flight 02: AWAS observations, modelled boundary layer*
*contribution, the modelled total mixing ratios for CH₃I, CHBr₃ and CH₂Br₂. The boundary layer and*
*background fractions means and standard deviations (in brackets) are given based on the*
*measurements and modelled values for the samples collected during the flight.*

| Altitude [km] | AWAS [ppt] | Modelled Boundary Layer Contribution [ppt] | Modelled Total Mixing Ratio [ppt] |
|---|---|---|---|
| *CH₃I* | | | |
| 17-18 | 0.06 (0.02) | 0.00 (0.00) | 0.06 (0.02) |
| 16-17 | 0.09 (0.03) | 0.00 (0.00) | 0.06 (0.02) |
| 15-16 | 0.17 (0.03) | 0.04 (0.04) | 0.12 (0.06) |
| 14-15 | 0.23 (0.09) | 0.17 (0.04) | 0.21 (0.08) |
| | | | |
| *CHBr₃* | | | |
| 17-18 | 0.34 (0.17) | 0.01 (0.00) | 0.29 (0.15) |
| 16-17 | 0.42 (0.11) | 0.03 (0.01) | 0.36 (0.14) |
| 15-16 | 0.55 (0.06) | 0.12 (0.07) | 0.48 (0.17) |
| 14-15 | 0.67 (0.10) | 0.35 (0.07) | 0.58 (0.13) |
| | | | |
| *CH₂Br₂* | | | |
| 17-18 | 0.72 (0.02) | 0.02 (0.01) | 0.71 (0.03) |
| 16-17 | 0.79 (0.07) | 0.06 (0.02) | 0.76 (0.06) |
| 15-16 | 0.83 (0.05) | 0.19 (0.09) | 0.78 (0.10) |
| 14-15 | 0.89 (0.05) | 0.46 (0.08) | 0.84 (0.12) |

| | *Boundary Layer fraction [%]* | *Background fraction [%]* |
|---|---|---|
| 17-18 | 2.1 (1.1) | 97.9 |
| 16-17 | 7.2 (2.7) | 92.8 |
| 15-16 | 22.9 (10.0) | 77.1 |
| 14-15 | 53.3 (9.0) | 46.7 |





*Table 3. ATTREX 2014 all flights: AWAS observations, modelled boundary layer contribution, the modelled total mixing ratios for $CH_3I$, $CHBr_3$ and $CH_2Br_2$. The boundary layer and background fractions are also given. Means and standard deviations (in brackets).*

| Altitude [km] | AWAS [ppt] | Modelled Boundary Layer Contribution [ppt] | Modelled Total Mixing Ratio [ppt] |
|---|---|---|---|
| *$CH_3I$* | | | |
| 17-18 | 0.04 (0.03) | 0.02 (0.03) | 0.07 (0.04) |
| 16-17 | 0.11 (0.10) | 0.04 (0.04) | 0.09 (0.05) |
| 15-16 | 0.16 (0.14) | 0.09 (0.07) | 0.15 (0.08) |
| 14-15 | 0.17 (0.14) | 0.15 (0.08) | 0.19 (0.11) |
| | | | |
| *$CHBr_3$* | | | |
| 17-18 | 0.33 (0.14) | 0.06 (0.06) | 0.32 (0.16) |
| 16-17 | 0.48 (0.13) | 0.12 (0.09) | 0.40 (0.17) |
| 15-16 | 0.54 (0.13) | 0.21 (0.12) | 0.50 (0.19) |
| 14-15 | 0.61 (0.13) | 0.31 (0.12) | 0.55 (0.16) |
| | | | |
| *$CH_2Br_2$* | | | |
| 17-18 | 0.73 (0.06) | 0.11 (0.09) | 0.73 (0.09) |
| 16-17 | 0.82 (0.08) | 0.19 (0.14) | 0.78 (0.15) |
| 15-16 | 0.84 (0.09) | 0.32 (0.16) | 0.80 (0.17) |
| 14-15 | 0.86 (0.07) | 0.44 (0.15) | 0.84 (0.17) |
| | | | |
| | *Boundary Layer fraction [%]* | *Background fraction [%]* | |
| 17-18 | 12.7 (10.9) | 87.3 | |
| 16-17 | 22.3 (16.0) | 77.7 | |
| 15-16 | 37.8 (18.8) | 62.2 | |
| 14-15 | 51.7 (16.1) | 48.3 | |






*Table 4. ATTREX 2013 all flights: AWAS observations, modelled boundary layer contribution, the*
*modelled total mixing ratios for CH₃I, CHBr₃ and CH₂Br₂. The boundary layer and background*
*fractions are also given. Means and standard deviations (in brackets).*

| Altitude [km] | AWAS [ppt] | Modelled Boundary Layer Contribution [ppt] | Modelled Total Mixing Ratio [ppt] |
|---|---|---|---|
| *$CH_3I$* | | | |
| 17-18 | 0.03 (0.02) | 0.00 (0.00) | 0.03 (0.01) |
| 16-17 | 0.03 (0.02) | 0.00 (0.00) | 0.03 (0.02) |
| 15-16 | 0.04 (0.02) | 0.01 (0.01) | 0.03 (0.03) |
| 14-15 | 0.04 (0.03) | 0.01 (0.01) | 0.05 (0.03) |
| | | | |
| *$CHBr_3$* | | | |
| 17-18 | 0.31 (0.10) | 0.01 (0.01) | 0.31 (0.09) |
| 16-17 | 0.39 (0.12) | 0.02 (0.02) | 0.35 (0.11) |
| 15-16 | 0.54 (0.15) | 0.04 (0.04) | 0.49 (0.16) |
| 14-15 | 0.53 (0.15) | 0.07 (0.05) | 0.53 (0.18) |
| | | | |
| *$CH_2Br_2$* | | | |
| 17-18 | 0.79 (0.08) | 0.02 (0.04) | 0.78 (0.07) |
| 16-17 | 0.83 (0.07) | 0.04 (0.04) | 0.81 (0.07) |
| 15-16 | 0.90 (0.07) | 0.07 (0.06) | 0.87 (0.10) |
| 14-15 | 0.91 (0.08) | 0.12 (0.09) | 0.89 (0.12) |
| | | | |
| | *Boundary Layer fraction [%]* | *Background fraction [%]* | |
| 17-18 | 1.9 (2.3) | 98.1 | |
| 16-17 | 4.7 (4.9) | 95.3 | |
| 15-16 | 9.8 (7.9) | 90.2 | |
| 14-15 | 14.7 (11.1) | 85.3 | |







*Table 5. Contribution from the very short-lived bromocarbons: CHBr$_3$ and CH$_2$Br$_2$ to the bromine in*
*the TTL as given by modelled estimates and AWAS observations for ATTREX 2014 and 2013.*
*[CHBr$_3$ ] and [CH$_2$Br$_2$] means are shown only.*

| Altitude [km] | [CHBr$_3$] [ppt] | [CH$_2$Br$_2$] [ppt] | Br from CHBr$_3$ [ppt] | Br from CH$_2$Br$_2$ [ppt] | Br-VSL$_{org}$ [ppt] |
|---|---|---|---|---|---|
| *ATTREX 2014* | | | | | |
| *NAME* | | | | | |
| 17-18 | 0.32 | 0.73 | 0.96 | 1.46 | 2.42 |
| 16-17 | 0.40 | 0.78 | 1.20 | 1.56 | 2.76 |
| 15-16 | 0.50 | 0.80 | 1.50 | 1.60 | 3.10 |
| 14-15 | 0.55 | 0.84 | 1.65 | 1.68 | 3.33 |
| *AWAS* | | | | | |
| 17-18 | 0.33 | 0.73 | 0.99 | 1.46 | 2.45 |
| 16-17 | 0.48 | 0.82 | 1.44 | 1.64 | 3.08 |
| 15-16 | 0.54 | 0.84 | 1.62 | 1.68 | 3.30 |
| 14-15 | 0.61 | 0.86 | 1.83 | 1.72 | 3.55 |
| *ATTREX 2013* | | | | | |
| *NAME* | | | | | |
| 17-18 | 0.31 | 0.78 | 0.93 | 1.56 | 2.49 |
| 16-17 | 0.35 | 0.81 | 1.05 | 1.62 | 2.67 |
| 15-16 | 0.49 | 0.87 | 1.47 | 1.74 | 3.21 |
| 14-15 | 0.53 | 0.89 | 1.59 | 1.78 | 3.37 |
| *AWAS* | | | | | |
| 17-18 | 0.31 | 0.79 | 0.93 | 1.58 | 2.51 |
| 16-17 | 0.39 | 0.83 | 1.17 | 1.66 | 2.83 |
| 15-16 | 0.54 | 0.90 | 1.62 | 1.80 | 3.42 |
| 14-15 | 0.53 | 0.91 | 1.59 | 1.82 | 3.41 |




**12 Figures**

**Figure 1:** Vertical distribution of CH₃I, CHBr₃ and CH₂Br₂ in the TTL, as measured during Research
Flight 02, ATTREX 2014: AWAS measurements along the flight track (left), observations grouped
into 1 km TTL segments (right, means (star symbols), standard deviations (coloured whiskers),
minimum, lower and upper quartiles, median and maximum (black box and whiskers)).

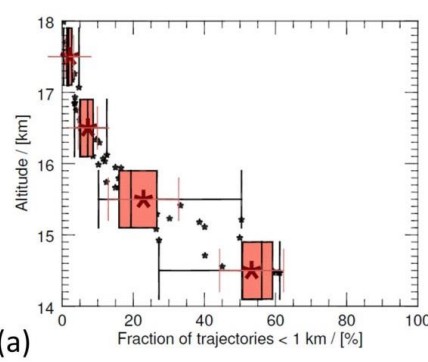
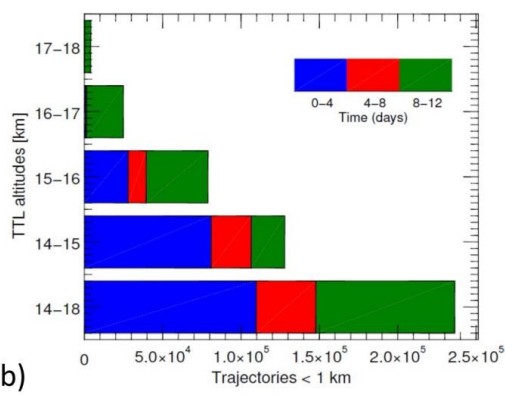


**Figure 2:** Vertical distribution of NAME 1 km fractions (the fractions which reach the boundary layer within 12 days - indicative of boundary layer air influence) in the TTL (2a, left). Distribution of transport times taken for the trajectories to first cross below 1 km (reach boundary layer) for all the NAME runs and the NAME runs grouped into 1 km TTL segments, Research Flight 02, ATTREX 2014 (2b, right).


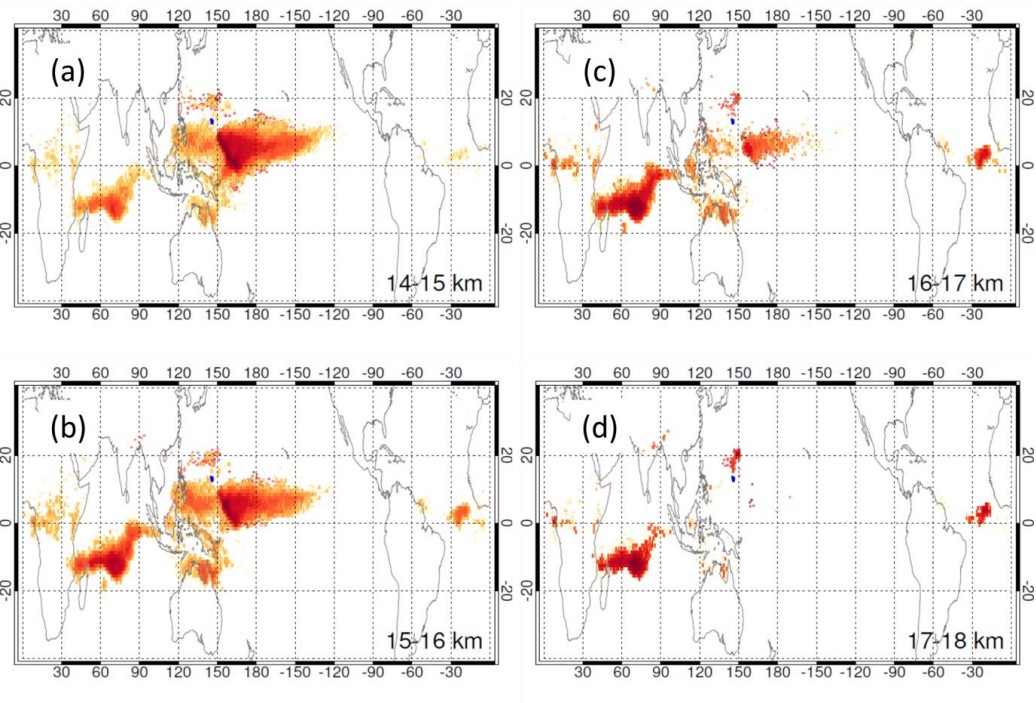






**Figure 3:** Crossing location distribution maps for all the NAME runs released from 4 1 km TTL
altitudes: 14-18 km. Strong influence of local boundary air is noted for a 14-15 km segment (lower
TTL), whereas the boundary air from remote locations dominates for a 17-18 km segment (upper
TTL), Research Flight 02, ATTREX 2014.

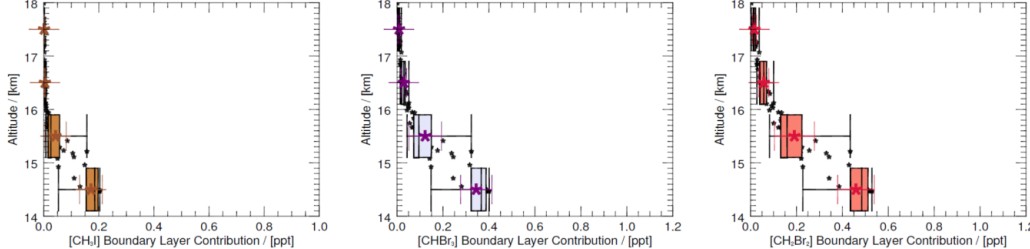


**Figure 4:** NAME modelled $CH_3I$, $CHBr_3$ and $CH_2Br_2$ boundary layer contribution to the TTL,
Research Flight 02, ATTREX 2014.

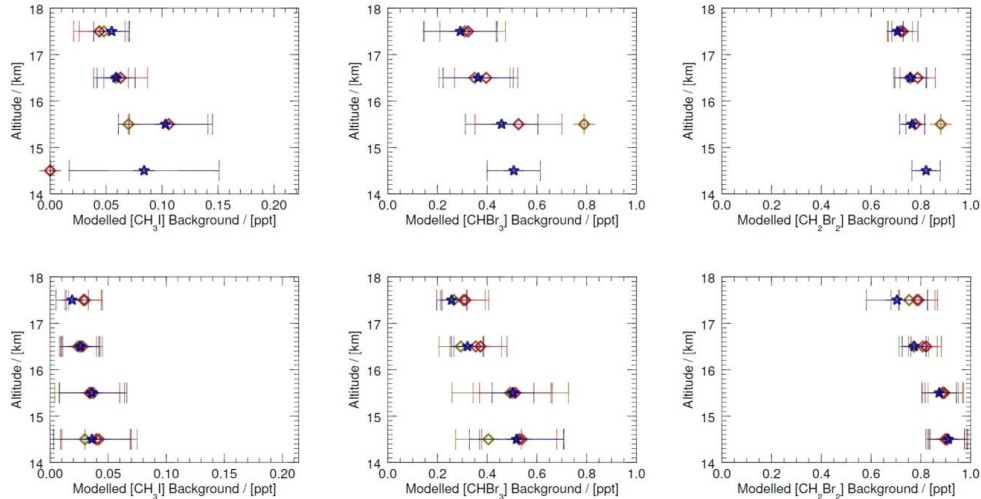


**Figure 5:** Background mixing ratios for $CH_3I$, $CHBr_3$ and $CH_2Br_2$ for all NAME runs for all flights
in ATTREX 2014 (top row) and ATTREX 2013 (bottom row). Little convective influence is
indicated by selecting means from NAME 1 km fractions of <1 (blue star), 5 (red diamond) and 10
(green diamond) %.



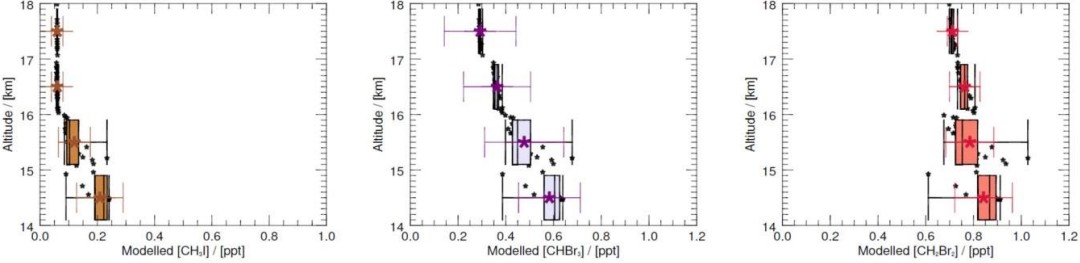


**Figure 6:** Vertical distribution of NAME modelled $CH_3I$, $CHBr_3$ and $CH_2Br_2$ (sums of boundary layer and background contribution) in the TTL for Research Flight 02, ATTREX 2014.


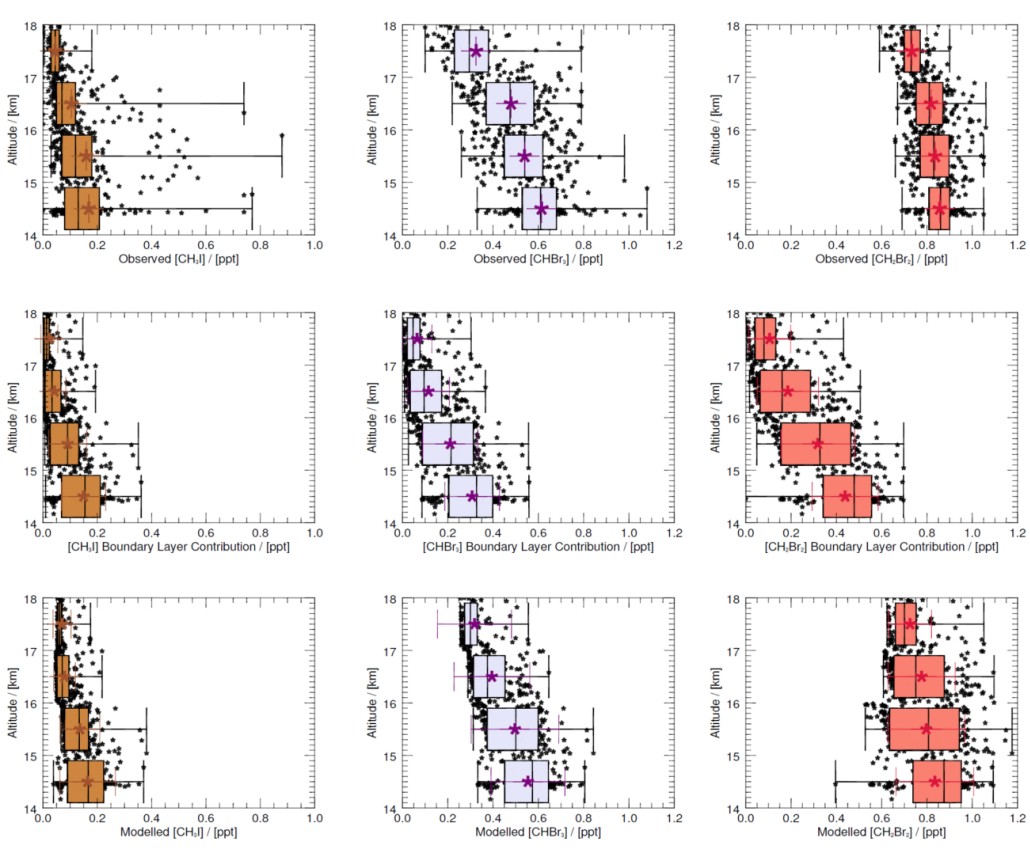





**Figure 7:** CH$_3$I, CHBr$_3$ and CH$_2$Br$_2$ vertical distribution in the TTL for ATTREX 2014 flights:
AWAS observations (top row), NAME modelled boundary layer contribution (middle row), and
NAME modelled sums of boundary layer and background contributions (bottom row).


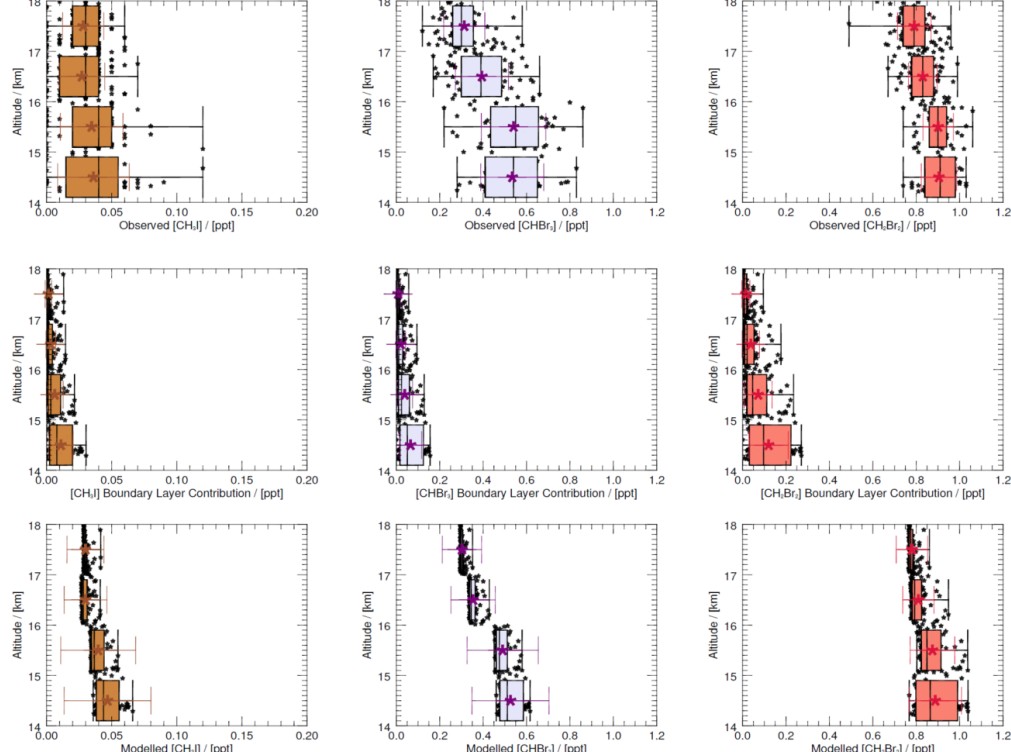


**Figure 8:** CH$_3$I, CHBr$_3$ and CH$_2$Br$_2$ vertical distribution in the TTL for ATTREX 2013 flights:
AWAS observations (top row), NAME modelled boundary layer contribution (middle row), and
NAME modelled sums of boundary layer and background contributions (bottom row).



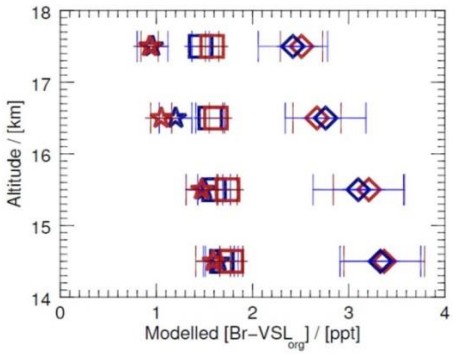 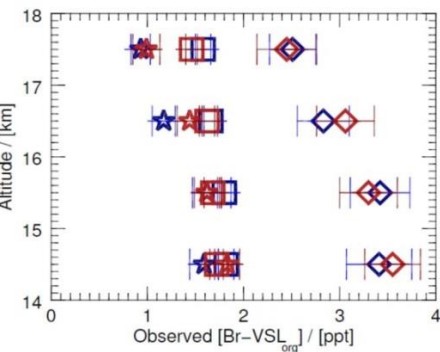


**Figure 9:** Contribution of CHBr$_3$ (star symbol) and CH$_2$Br$_2$ (square symbol) to the bromine budget in the TTL, inferred from the NAME modelled estimates (left) and AWAS observations (right); separately ATTREX 2014 (red) and 2013 (blue). Star and square symbols represent the bromine atomicity products from CHBr$_3$ and CH$_2$Br$_2$, respectively. Diamonds show the bromine contribution from the VSL bromocarbons in the TTL (as a sum of the CHBr$_3$ and CH$_2$Br$_2$ bromine atomicity products).

776