# Peer review of "Transport of short-lived halocarbons to the stratosphere over the Pacific Ocean."

_Atmospheric Chemistry and Physics, 2018_

## Referee Comment (RC1) · Anonymous Referee #1 · 2 Nov 2018

This work builds on the 2014 joint CAST/CONTRAST/ATTREX missions where VSLS ($CHBr_3$, $CH_2Br_2$, $CH_3I$) measurements were made in the tropical West Pacific. Here, the NAME model is used to compute back trajectories from the VSLS measurement location/times and to determine the fraction of released particles that crossed the boundary layer in the preceding 12 days. With this information, the authors estimate the influence of the boundary layer on VSLS mixing ratios (during the campaign period) throughout the vertical extent of the TTL and compare model estimates of VSLS to the actual observations.

I find the description of the method reasonably straight forward to follow and the paper interesting. The results are certainly within the scope of ACP and, in addition to describing a method for interpreting measurements, the paper contributes some analysis

on differences in measured VSLS concentrations between ATTREX 2013 (W Pacific) and 2014 (E Pacific). My main concerns (outlined below) are on the use of assumed chemical decay times and on some aspects of the manuscript presentation. Both issues could be addressed readily, and I recommend the paper for publication.

(1) The authors use constant chemical decay lifetimes of 15 days and 94 days for CHBr3 and CH2Br2, respectively, based on the boundary layer estimates given by Carpenter et al. (2014). Can the use of a fixed lifetime be justified given that local lifetimes of the above compounds are known to vary substantially between the surface and in the TTL (e.g. Hossaini et al., 2010, Liang et al., 2010)? These references show a much longer TTL CH2Br2 lifetime than 94 days, for example. Accounting for photochemical removal along trajectories is important and the authors should comment on how sensitive their findings (e.g. boundary layer contributions in the TTL) are to the lifetime assumptions.

(2) The presentation of the manuscript could be improved in several places. Specific suggestions are given below. Additionally, throughout the manuscript the authors should consider whether the citations given are the most appropriate to the points made in the text. An example is on Line 50 where the point is that VSLS are emitted from the ocean and have natural sources. Given that, citations to modelling work looking at impacts of iodine/bromine chemistry (Solomon, Vogt, Salawitch, Saiz-Lopez) seem somewhat out of place. More appropriate and recent references would be, for example:

Hepach, H., et al. Biogenic halocarbons from the Peruvian upwelling region as tropospheric halogen source, Atmos. Chem. Phys., 16, 12219-12237, 2016.

Hepach, H., et al. Halocarbon emissions and sources in the equatorial Atlantic Cold Tongue, Biogeosciences, 12, 6369-6387, 2015.

Yang, G. et al. Spatio-temporal variations of sea surface halocarbon concentrations and fluxes from southern Yellow Sea, Biogeochemistry, 37 121(2), 369-388, 2014.

[Figure]

I suggest the authors thoroughly proof the paper for similar instances and areas where readability could be improved.

Specific comments: -Line 38: The issue of "continued depletion in the lower stratosphere" is debatable. Mid-latitude and tropical ozone in this region is strongly influenced by transport and much of the apparent downward trend reported by Ball et al. appears to be have been reversed in 2017, as shown by Chipperfield et al. (2018). I would encourage the authors to amend this sentence to a more precise one.

Chipperfield, M. P., Dhomse, S., Hossaini, R., Feng, W., Santee, M. L., Weber, M., et al. (2018). On the cause of recent variations in lower stratospheric ozone. Geophysical Research Letters, 45, 5718–5726. https://doi.org/10.1029/2018GL078071

-Line 47: on first appearance spell out the name of these compounds: i.e. methyl iodide (CH3I), bromoform (CHBr3) and dibromoethane (CH2Br2).

-Line 52: Is there a reason why specifically 12 days is chosen? In the Discussion (line 461), it is noted that longer periods are tested but the details are very vague. I would state earlier on in the manuscript that sensitivity tests were performed and be more quantitative on what was found.

-Line 82: "east" — "East"

-In Section 2.1 it would be useful to indicate the altitude limits of the various aircrafts. Related to this, it would help the reader to know how the TTL is being defined up front.

-Line 140: The citation to Jones et al. should probably appear directly after NAME.

-Line 215: Should "Research Flights" have capital letters?

-Line 217: "very short lived brominated substances" could be deleted

-Line 222: Starting a sentence with this number is a bit odd. Consider rewording or spelling out the number.

[Figure]

-Line 223: "TTL" — "the TTL"

-Line 248: "western and central" — "Western and Central"

-Line 387: define MJO

-Figure 1 caption: I recommend reworking as brackets within brackets looks odd here.

-Figure 2 caption: What are the black symbols? Should also indicate if box and whiskers are the same as Figure 1.

-Figure 4: A reduced x-axis scale for each species would improve readability of the data.

---

## Referee Comment (RC2) · Anonymous Referee #2 · 23 Nov 2018

This paper about "Transport of short-lived halocarbons to the stratosphere over the Pacific Ocean" by Michal T. Filus et al. reports about transport of VSLS above the West Pacific using the Lagrangian model NAME and new aircraft observations from the joint CAST, CONTRAST, ATREX campaign in Jan-Mar 2014. The authors use an improved NAME version which includes a convection scheme. This methodology has been applied to many VSLS transport studies before and is a common procedure in the community. However, as the authors investigate the VSLS transport from the boundary layer to the stratosphere comparing it with a new aircraft campaign and a further developed model version of NAME I believe it can fulfil the criteria to be published in ACP after carefully revising the paper including better specifying the new perspective of your study, the state of the art and background in this field and a thorough discussion of the

study uncertainties. See my specific comments below.

I) What is really new in your study? To use a Langrangian dispersal model including a convection scheme is nothing new in this field. Next there were several studies including the VSLS contribution to the stratosphere for CAST/CONTRAST/ATREX. Thus, I suggest to think carefully about what is different and thus really new compared to example i) the old NAME VSLS studies, ii) the FLEXPART model and VSLS studies (including a convection scheme) and iii) compared to other VSLS CAST/CONTRAST/ATREX studies, see Wales et al 2018 JGR. This new perspective should be clearer addressed in the introduction and could be added to the discussion of your results.

Line 85: "using a new Lagrangian methodology" I suggest deleting "new" as it is not a new method.

II) What is the state of the art in this research field? Here it seems to me that you are mainly referring to new recent studies and did not go back to the original literature. One example is the citation of the oceanic source of VSLS where you mainly cite VSLS modelling studies, which should be original biogeochemical oceanographic articles such as e.g. Carpenter et al., 1999; Moore and Zafiriou, 1994; Quack and Wallace, 2003 among others. Be aware of the different VSLS components which have different oceanic sources and thus will request different articles to cite. Overall, I suggest to carefully going through all references again citing also the specific original work instead of large selections of recent, maybe randomly chosen, papers.

III) Discuss the uncertainties of your VSLS transport calculations: What is the uncertainty due to the model and meteorology used, transport processes (e.g. BL vs convection scheme), using constant VSLS life times? (see Hossaini et al 2010, Fuhlbrügge et al 2016). - How good is the "Meteorological Office's Unified Model" meteorology compared to the actual observed meteorology? Here, I refer to observed convection events and winds. How much does the use of this specific meteorology fields affect your results? -Btw, what kind of model is it (operational, assimilation or?) - If I under-
stand it correctly you use constant VSLS lifetimes. Is this appropriate (see Hossaini et al 2010, Liang et al 2010) and what would you expect the results to be using vertical varying lifetimes? I assume you cannot change and add new runs anymore, but you should add a clear and thorough discussion here at least! -How different are your NAME results compared to other transport model studies? (e.g. Fig. 3)?

Figures and text: Thoroughly revise your figures quality. Often the labelling is too small and unreadable on my print out. How about adding a line to your profiles? The main text and references still need revision and editorial help (typos).

---

## Author Comment (AC1) · 8 Aug 2019

This work builds on the 2014 joint CAST/CONTRAST/ATTREX missions where VSLS (CHBr$_3$, CH$_2$Br$_2$, CH$_3$I) measurements were made in the tropical West Pacific. Here, the NAME model is used to compute back trajectories from the VSLS measurement location/times and to determine the fraction of released particles that crossed the boundary layer in the preceding 12 days. With this information, the authors estimate the influence of the boundary layer on VSLS mixing ratios (during the campaign period) throughout the vertical extent of the TTL on differences in measured VSLS concentrations between ATTREX 2013 (W Pacific) and 2014 (E Pacific).

My main concerns (outlined below) are on the use of assumed chemical decay times and on some aspects of the manuscript presentation. Both issues could be addressed readily, and I recommend the paper for publication.

We thank both reviewers for their constructive comments. In our opinion, these have resulted in an improved manuscript.

(1) The authors use constant chemical decay lifetimes of 15 days and 94 days for CHBr$_3$ and CH$_2$Br$_2$, respectively, based on the boundary layer estimates given by Carpenter et al. (2014). Can the use of a fixed lifetime be justified given that local lifetimes of the above compounds are known to vary substantially between the surface and in the TTL (e.g. Hossaini et al., 2010, Liang et al., 2010)? These references show a much longer TTL CH$_2$Br$_2$ lifetime than 94 days, for example. Accounting for photochemical removal along trajectories is important and the authors should comment on how sensitive their findings (e.g. boundary layer contributions in the TTL) are to the lifetime assumptions.

The following text has been added as a new subsection at the end of section 2.

**"2.3.3 The effect of assuming constant lifetimes**

The lifetimes of the halocarbons are not the same in the boundary layer and the TTL (Carpenter et al, 2014). The assumption of constant lifetime in a 12 day trajectory is evaluated by calculating the difference between idealised trajectories which had 2, 4, 6, 8, and 10 days in the boundary layer and 10, 8, 6, 4, and 2 days in the upper troposphere. Lifetimes for the boundary layer and for the upper troposphere for each gas were taken from Carpenter et al. (2014). (Lifetimes for higher altitudes are not available therein). The difference found between the two extreme cases are 6% (CHBr$_3$), 3% (CH$_2$Br$_2$) and 25% (CH$_3$I). The assumption is thus valid for the two brominated species.

This assumption is more robust than it might seem at first glance. The boundary layer fraction is calculated using 12 day trajectories in which there is little loss of CH$_2$Br$_2$ whether a lifetime of 94 or 150 days is taken. The most important factor in determining the amount lofted into the TTL is thus the original mixing ratio which is only slightly modulated by the chemical loss in 12 days. The longer lifetime is absorbed implicitly taken into account in the background contribution. The same arguments apply for CHBr$_3$, though the effect is a bit larger. The largest difference is seen for CH$_3$I. However, the difference matters much less for CH$_3$I because only 4-5% remains after the full 12 days which is much smaller than the uncertainties in this analysis so that much shorter trajectories are used to validate the new convection scheme."

(2) The presentation of the manuscript could be improved in several places. Specific suggestions are given below. Additionally, throughout the manuscript the authors should consider whether the citations given are the most appropriate to the points made in the text. An example is on Line 50 where the point is that

VSLS are emitted from the ocean and have natural sources. Given that, citations to modelling work looking at impacts of iodine/bromine chemistry (Solomon, Vogt, Salawitch, Saiz-Lopez) seem somewhat out of place. More appropriate and recent references would be, for example:

Hepach, H., et al. Biogenic halocarbons from the Peruvian upwelling region as tropospheric halogen source, Atmos. Chem. Phys., 16, 12219-12237, 2016.

Hepach, H., et al. Halocarbon emissions and sources in the equatorial Atlantic Cold Tongue, Biogeosciences, 12, 6369-6387, 2015.

Yang, G. et al. Spatio-temporal variations of sea surface halocarbon concentrations and fluxes from southern Yellow Sea, Biogeochemistry, 37 121(2), 369-388, 2014.

We have also read through it carefully and tried to improve the clarity. The point about the referencing is taken and we have added these and some other, more relevant references to the manuscript with that in mind.

I suggest the authors thoroughly proof the paper for similar instances and areas where readability could be improved.

We have carefully read through the papers with a view to making it clearer to the reader.

Specific comments:

Line 38: The issue of "continued depletion in the lower stratosphere" is debatable. Mid-latitude and tropical ozone in this region is strongly influenced by transport and much of the apparent downward trend reported by Ball et al. appears to be have been reversed in 2017, as shown by Chipperfield et al. (2018). I would encourage the authors to amend this sentence to a more precise one. Chipperfield, M. P., Dhomse, S., Hossaini, R., Feng, W., Santee, M. L., Weber, M., et al. (2018). On the cause of recent variations in lower stratospheric ozone. Geophysical Research Letters, 45, 5718–5726. https://doi.org/10.1029/2018GL078071

We have added the reference to Chipperfield et al (2018). However we note the recent publication of Ball et al (2019) in ACPD and think the jury is still out. We have changed 'depletion' to 'possible reduction' due to the likelihood of its origin as being dynamic.

Line 47: on first appearance spell out the name of these compounds: i.e. methyl iodide (CH3I), bromoform (CHBr3) and dibromoethane (CH2Br2).

Names of these compounds have been spelled out.

Line 52: Is there a reason why specifically 12 days is chosen? In the Discussion (line 461), it is noted that longer periods are tested but the details are very vague. I would state earlier on in the manuscript that sensitivity tests were performed and be more quantitative on what was found.

See above

Line 82: "east" — "East"

This has been corrected.

In Section 2.1 it would be useful to indicate the altitude limits of the various aircrafts. Related to this, it would help the reader to know how the TTL is being defined up front.

Agreed. Altitude limits of the various aircraft have been added:

CAST BAe-164        0-8 km

Gulfstream V          1-14 km

Global Hawk       13-19 km

-Line 140: The citation to Jones et al. should probably appear directly after NAME.

Agreed – this citation appears directly after NAME.

Line 215: Should "Research Flights" have capital letters?

We have switched to lower case, except in section and caption titles.

Line 217: "very short lived brominated substances" could be deleted

Agreed. These words have been deleted.

Line 222: Starting a sentence with this number is a bit odd. Consider rewording or spelling out the number.

Agreed. This number has been spelled out.

Line 223: "TTL" — "the TTL"

Agreed.

Line 248: "western and central" — "Western and Central"

Agreed.

Line 387: define MJO

Agreed, the MJO has now been defined as Madden-Julian Oscillation.

Figure 1 caption: I recommend reworking as brackets within brackets looks odd here.

Brackets have now been removed and replaced with hyphens.

Figure 2 caption: What are the black symbols? Should also indicate if box and whiskers are the same as Figure 1.

Agreed, black symbols are the same as Figure 1 and represent measurements.

Figure 4: A reduced x-axis scale for each species would improve readability of the data.

We have used this scale to be consistent and for easier comparison of Figure 4 with Figures 1 and 6. We fully understand the suggestion and the reviewer's intention to improve readability of the data by reducing x-axis scale but we would prefer to keep it unchanged to help readers compare the data between multiple figures. We are happy to accept the editor's judgement on this.

---

## Author Comment (AC2) · 8 Aug 2019

This paper about "Transport of short-lived halocarbons to the stratosphere over the Pacific Ocean" by Michal T. Filus et al. reports about transport of VSLS above the West Pacific using the Lagrangian model NAME and new aircraft observations from the joint CAST, CONTRAST, ATREX campaign in Jan-Mar 2014. The authors use an improved NAME version which includes a convection scheme. This methodology has been applied to many VSLS transport studies before and is a common procedure in the community. However, as the authors investigate the VSLS transport from the boundary layer to the stratosphere comparing it with a new aircraft campaign and a further developed model version of NAME I believe it can fulfil the criteria to be published in ACP after carefully revising the paper including better specifying the new perspective of your study, the state of the art and background in this field and a thorough discussion of the study uncertainties.

We thank both reviewers for their constructive comments. In our opinion, these have resulted in an improved manuscript.

See my specific comments below.

I)      What is really new in your study? To use a Langrangian dispersal model including a convection scheme is nothing new in this field. Next there were several studies including the VSLS contribution to the stratosphere for CAST/CONTRAST/ATREX. Thus, I suggest to think carefully about what is different and thus really new compared to example (i) the old NAME VSLS studies, ii) the FLEXPART model and VSLS studies (including a convection scheme) and iii) compared to other VSLS CAST/CONTRAST/ATREX studies, see Wales et al 2018 JGR. This new perspective should be clearer addressed in the introduction and could be added to the discussion of your results.

The main new aspects of this study are:

(a)   the validation and use of an improved convection scheme for use with the NAME trajectory model. The previous scheme was reasonable for convection at mid-latitudes but was far too weak to represent the stronger tropical convection. Comparison with the extensive CH3I measurements made in this campaign provides good support for its use in modelling transport in tropical convective systems.

(b)   The old convective scheme was used in the earlier study by Ashfold et al (2012) using the East Pacific measurements, so the new scheme represents a considerable improvement which found reasonable agreement only up to and including the level of maximum convective outflow.

(c)   We have extended the approach used by Ashfold et al (2012) so that VSLS mixing ratios can be assigned to contributions from the boundary layer and from the 'background' TTL.

(d)   The FLEXPART studies focussed on transport up to the level of maximum convective outflow during the SHIVA campaign based in Malaysian Borneo and had a less complete set of measurements to compare with. The surface concentrations and strength of convection over the South China Sea are different to those over the West Pacific in Jan-Mar.

(e)   The conclusions of the Wales analysis are based on the Eulerian 3D CAM-chem-SD model while ours are based purely on a trajectory-based approach. The agreement is good.

(f)   We compare results from 2 years (2013 and 2014)

We have changed the introduction a bit to lay the groundwork for a summary of these points in the Summary and Discussion.

Line 85: "using a new Lagrangian methodology" I suggest deleting "new" as it is not a new method.

This has now been deleted and has been replaced by 'updated' in several places. The 'new' aspects of the overall methodology we were referring to were (a) it is a measurement-based way of the quantifying boundary layer and background contributions to brominated VSLS budget in the TTL; and (ii) using and testing with CH3I the improved parameterisation for deep convection developed in the NAME model).

II)     What is the state of the art in this research field? Here it seems to me that you are mainly referring to new recent studies and did not go back to the original literature. One example is the citation of the oceanic source of VSLS where you mainly cite VSLS modelling studies, which should be original biogeochemical oceanographic articles such as e.g. Carpenter et al., 1999; Moore and Zafiriou, 1994; Quack and Wallace, 2003 among others. Be aware of the different VSLS components which have different oceanic sources and thus will request different articles to cite. Overall, I suggest to carefully going through all references again citing also the specific original work instead of large selections of recent, maybe randomly chosen, papers.

We have improved the discussion on the state of the art in the introduction and changed some of the references.

III)    Discuss the uncertainties of your VSLS transport calculations:

What is the uncertainty due to the model and meteorology used, transport processes (e.g. BL vs convection scheme), using constant VSLS life times? (see Hossaini et al 2010, Fuhlbrügge et al 2016). How good is the "Meteorological Office's Unified Model" meteorology compared to the actual observed meteorology? Here, I refer to observed convection events and winds. How much does the use of this specific meteorology fields affect your results?

The uncertainty is likely to be dominated by the errors in the convection. The boundary layer dispersion scheme is likely to be unimportant as we only track the parcels back until they reach within 1km of the surface. Also the winds from the Unified Model (UM) are expected to be accurate, partly because they are from analyses rather than from forecasts, but also because the UM is among the best operational forecast models – see e.g.  https://apps.ecmwf.int/wmolcdnv/ . [It is hard to quantify the errors though, because the analysis is, by definition, our best estimate of the truth, obtained by assimilating a range of observations which themselves have errors. Indeed the analysed winds are often used as the benchmark against which to assess forecasts.]

Convection is difficult to predict well, especially with a large scale global model where the convection is sub grid scale. Fig 5 in Geosci. Model Dev. vol. 12, p. 1909 (2019) shows climatological cloud over the Pacific warm pool from the global UM compared with Calipso satellite data. This shows reasonable predictions, although with the convection not being quite deep enough. This is consistent with the comparison between model and aircraft data. We expect the errors for individual convective events to be significant, but the upper troposphere concentrations will depend on a number of convective events and we are considering a range of flights and measurements locations, which we hope makes the conclusions on general behaviour robust. Again the consistency between model and aircraft data supports this. One could attempt a more detailed estimate of

errors by using data from a range of models and from ensemble prediction systems, but that would be another project.

We have added some discussion of these issues to the Summary and Discussion section.

-Btw, what kind of model is it (operational, assimilation or?)

We used operational analyses from the UK Meteorological Office in this study. This has been clarified in the text. Operational forecasts were used during the campaign to assist with planning (Harris et al., BAMS, 2017), but are not considered here.

If I understand it correctly you use constant VSLS lifetimes. Is this appropriate (see Hossaini et al 2010, Liang et al 2010) and what would you expect the results to be using vertical varying lifetimes? I assume you cannot change and add new runs anymore, but you should add a clear and thorough discussion here at least!

Please see response to reviewer 1.

How different are your NAME results compared to other transport model studies? (e.g. Fig. 3)?

A comparison of our results with those from Wales et al (2018) has been added at the end of Section 5. There were existing references to Navarro (2015) which included a comparison with the WACCM model and to Butler et al (updated to 2018). Feng et al (2018) is relevant and uses the same observations, but focuses on ocean-atmosphere fluxes so is not comparable. We are not aware of other papers. References to studies of regions outside the Western Pacific are made elsewhere (e.g. Tegtmeier et al 2012, 2013 and Fuhlbrügge et al 2016.

Figures and text: Thoroughly revise your figures quality. Often the labelling is too small and unreadable on my print out. How about adding a line to your profiles?

The figure quality has been improved as suggested, We prefer not to add a line to the plots of the vertical profiles as we think the information is easier for the reader to grasp without it. We are happy to consider further suggestions.

The main text and references still need revision and editorial help (typos).

We have gone through the main text and references carefully.

---

## Author Comment (AC3) · 8 Aug 2019

The comment was uploaded in the form of a supplement:
https://www.atmos-chem-phys-discuss.net/acp-2018-640/acp-2018-640-AC3-supplement.pdf
* * *

---

## Author Response (AR2)

[revised manuscript text omitted]

Different studies focussed on transport up to the level of maximum convective outflow, including the ones where FLEXPART chemistry-transport model is applied, during the SHIVA campaign based in Malaysian Borneo. The surface concentrations and strength of convection over the South China Sea are different to those over the West Pacific in January – March (winter). Another more recent study by Wales et al., 2018 is based on the Eulerian 3D CAM-chem-SD model while this study is based purely on a trajectory-based approach. The agreement between these two studies is good.

Even though this methodology has been applied to many VSLS transport studies before and is a common procedure in the research community, we investigate the VSLS transport from the boundary layer to the stratosphere comparing it with a new multi-aircraft campaign [below but better phrased] and a further developed model version of NAME with improved convection scheme. It is one of the first studies in which we have combined atmospheric measurements of the entire troposphere and lower stratosphere in the West Pacific region in 2014, and the UK NAME Lagrangian particle dispersion model with improved parameterisation scheme for simulating displacement of particles due to convective motions, to quantify mixing ratios for $CH_3I$, $CHBr_3$ and $CH_2Br_2$ and their estimated contributions from the boundary layer and the background. Firstly, our methodology for quantifying mixing ratios of $CH_3I$ works well as modelled estimates were in good agreement with ATTREX measurements in the TTL. This study also showed that the boundary layer air is the sole source of $CH_3I$ in the upper troposphere – lower stratosphere in the region of deep and frequent convective activity. A bespoke good agreement between modelled and measured $CH_3I$ mixing ratios in the upper troposphere and the TTL makes us confident about the good performance of the improved parameterisation scheme for displacement of particles as a result of deep convection. This methodology, with validated convection scheme for $CH_3I$, was further applied to quantify mixing ratios of $CHBr_3$ and $CH_2Br_2$ in the TTL. As these compounds are longer lived than $CH_3I$, the boundary layer contribution estimates tend to have less role, with the challenge of estimating the background contribution estimate in a confident manner. The agreement between modelled and measured $CHBr_3$ mixing ratios was good, and for $CH_2Br_2$ satisfactory, and for both within the reported literature values. We are confident that our methodology for quantifying boundary layer contribution of $CH_3I$, $CHBr_3$ and $CH_2Br_2$ gives good agreement with measured data, and slightly less confident on the estimates of background contribution, particularly for $CH_2Br_2$. We would like to further test our methodology by applying it to quantify modelled mixing ratios of short lived bromocarbons and iodocarbons for any future campaigns that feature source receptor measurements being taken at the same time and region.

[revised manuscript text omitted]